# A Novel Treatment Strategy by Natural Products in NLRP3 Inflammasome-Mediated Neuroinflammation in Alzheimer’s and Parkinson’s Disease

**DOI:** 10.3390/ijms22031324

**Published:** 2021-01-29

**Authors:** Jun Ho Lee, Hong Jun Kim, Jong Uk Kim, Tae Han Yook, Kyeong Han Kim, Joo Young Lee, Gabsik Yang

**Affiliations:** 1College of Korea Medicine, Woosuk University, Jeonju-si, Jeollabuk-do 54986, Korea; celtece@daum.net (J.H.L.); kimboncho@woosuk.ac.kr (H.J.K.); ju1110@hanmail.net (J.U.K.); nasiss@naver.com (T.H.Y.); solip922@hanmail.net (K.H.K.); 2BK21plus Team, College of Pharmacy, The Catholic University of Korea, Bucheon 14662, Korea; joolee@catholic.ac.kr

**Keywords:** NLRP3 inflammasome, Alzheimer’s disease, Parkinson’s disease, natural products

## Abstract

Alzheimer’s disease (AD) and Parkinson’s disease (PD) are the most common neurodegenerative diseases. Many studies have demonstrated that the release of NLRP3 inflammasome-mediated proinflammatory cytokines by the excessive activation of microglia is associated with the pathogenesis of AD and PD and suggested that the NLRP3 inflammasome plays an important role in AD and PD development. In both diseases, various stimuli, such as Aβ and α-synuclein, accelerate the formation of the NLRP3 inflammasome in microglia and induce pyroptosis through the expression of interleukin (IL)-1β, caspase-1, etc., where neuroinflammation contributes to gradual progression and deterioration. However, despite intensive research, the exact function and regulation of the NLRP3 inflammasome has not yet been clearly identified. Moreover, there have not yet been any experiments of clinical use, although many studies have recently been conducted to improve treatment of inflammatory diseases using various inhibitors for NLRP3 inflammasome pathways. However, recent studies have reported that various natural products show improvement effects in the in vivo models of AD and PD through the regulation of NLRP3 inflammasome assembly. Therefore, the present review provides an overview of natural extraction studies aimed at the prevention or treatment of NLRP3 inflammasome-mediated neurological disorders. It is suggested that the discovery and development of these various natural products could be a potential strategy for NLRP3 inflammasome-mediated AD and PD treatment.

## 1. Introduction

Neuroinflammation, an inflammatory response that occurs in the damaged central nervous system, is an important factor in neurodegeneration. Various types of cells, including microglia, astrocytes, and macrophages, are involved in this complicated inflammatory process. Microglia activated by pathological triggers migrate to the location of the injury and stimulate an innate immune response, including the production of pro-inflammatory cytokines, resulting in neuronal death [1,2]. In addition, inflammatory mediators produced by activated microglia promote astrogliosis, which in turn amplifies the inflammatory response by the additional secretion of inflammatory mediators. Persistent inflammation that occurs during the pathogenesis of various neurological disorders in the central nervous system is mediated by activation of the NLRP3 inflammasome, a key innate immune sensor that is associated with the progression of several neurological disorders, including Alzheimer’s disease (AD), Parkinson’s disease (PD), traumatic brain injury, stroke, depression, and multiple sclerosis [3,4].

Currently, there are only five types of medicine for AD treatment (four types of cholinesterase inhibitor and one type of N-methyl-D-aspartate receptor blocker), and 13 types of medicine for PD treatment (two types of dopaminergic receptor blocker, five types of dopamine agonist, three types of monoamine oxidase inhibitor, three types of catechol O-methyltransferase inhibitor) which are approved by the U.S. Food and Drug Administration. However, these treatments that have been developed so far only slightly delay the progression of the disease and have limited efficacy for early-stage patients, lack an efficient drug delivery system, and have various effects [5,6].

In recent years, the drug mechanisms of the developed therapeutic agents have been expanding into inflammation, neuroprotection, vascular factors, metabolism, and neuro-regeneration [7]. The development of a natural product-derived therapeutic agent can avoid the side effects of existing medicines and enable a wide range of drug mechanisms to be found. Therefore, in this review, we summarize the contribution of the mechanism of neuroinflammation contributes to AD and PD development, and then describe the activation of the NLRP3 inflammasome and its pathogenic role in AD and PD. In addition, we attempt to update the current knowledge of the neuroprotective effects of natural products on AD and PD treatment through regulation of the NLRP3 inflammasome. Therefore, this review attempts to provide a new perspective and potential strategy for reducing the development and progression of neurological disorders through the regulation of natural products in the NLRP3 inflammasome.

## 2. Mechanism of NLRP3 Inflammasome Activation and Regulation

The NLRP3 inflammasome is the most recently studied inflammatory control complex (Figure 1). It is activated by various stimuli and it contributes to the pathology of inflammatory diseases. Despite extensive investigation, the mechanism of NLRP3 inflammasome activation remains unclear. However, it is known that NLRP3 inflammasome activation requires two signaling steps provided by several exogenous and endogenous activators. The first signal (priming signal) is the nuclear factor kappa B (NF-κB)-dependent transcription of NLRP3 and pro-IL-1β, which is triggered by the binding of the Toll-like receptor (TLR)4 ligand lipopolysaccharide (LPS) to the receptor [8,9,10]. The signal then promotes the expression of inflammasome components, including NLRP3, procaspase-1, and pro-IL-1β (Figure 1). The first signal is initiated by TLR4 and then relayed by relevant adapter molecules, including myeloid differentiation factor 88 (MyD88), IRAK1, and IRAK4, without the requirement for new protein synthesis [11,12,13]. The second signal (activation signal), involved in the assembly and activation of the NLRP3 complex, is induced by extracellular ATP, certain bacterial toxins, and various types of crystalline and particulate matter. This signal subsequently produces active cleaved caspase-1. When microglia are activated by the NLRP3 activator as part of the second signal, NLRP3 undergoes oligomerization through homologous NACHT domain interactions, which recruits the PYD domain to interact with the PYD of ASC that triggers ASC fibrillar binding. Next, the ASC aggregate recruits CARD, which interacts with the CARD domain of procaspase-1 to promote caspase-1 activation and the subsequent polymerization of ASC fibrils into a large fibrous protein complex called ASC speck [14,15]. Clustered procaspase-1 mediates self-cleavage and activation in the form of activated caspase-1, which cleaves the precursors of IL-1β and IL-18 to produce activated forms of IL-1β and IL-18 (Figure 1), triggering an inflammatory response and pyroptosis [16,17]. In this process, the post-translational modifications of NLRP3, such as deubiquitination and phosphorylation, are necessary for the assembly and activation of the NLRP3 inflammasome (second signal) [12,18]. However, the exact mechanism by which NLRP3 inflammasome assembly is promoted is still unknown. When the activation of NLRP3 is triggered, the formation of ASC speck is considered an upstream indication of NLRP3 activation [19].

In addition, potassium efflux, mitochondrial dysfunction, reactive oxygen species (ROS), and lysosome destruction were also identified as higher signaling events for NLRP3 inflammasome activation. Most NLRP3 stimuli can cause the outflow of potassium from cells. Conversely, the inhibition of potassium efflux by high extracellular potassium concentration blocks most stimulus-induced NLRP3 inflammasome activation. Thus, potassium efflux is a major upstream signaling event for NLRP3 inflammasome activation (Figure 1). In contrast, the role of mitochondria and ROS in NLRP3 inflammasome activation remains controversial. Numerous studies support the role of mitochondrial dysfunction and associated ROS in NLRP3 inflammasome activation. However, other studies have shown that ROS is only required to prime the NLRP3 inflammasome and that mitochondrial dysfunction is indispensable for NLRP3 inflammasome activation. In addition, lysosome damage is only required for NLRP3 inflammasome activation by particulate matter. Several studies have suggested that members of the cathepsin family released from damaged lysosomes mediate NLRP3 inflammasome activation (Figure 1). In addition to these upstream events, many NLRP3 interacting partners and post-translational variants of NLRP3 regulate NLRP3 inflammasome activation [20].

## 3. Role of NLRP3 Inflammasome in AD

NLRP3 inflammasome activation has been implicated in the pathogenesis of neurodegenerative diseases, especially AD and PD. The deposition of misfolded amyloid-β (Aβ) in the brain is a major pathological event in AD. Recent studies have shown that NLRP3 inflammasome-mediated neuroinflammation plays an important role in the pathogenesis of AD [21]. The secretion of IL-1β by Aβ was dependent on NLRP3, ASC, and caspase-1 activities and required cathepsin B released from damaged lysosomes. Furthermore, fewer microglia and mononuclear phagocytes were recruited to Aβ in the brains of ASC and caspase-1 knockout mice (IL-1 receptor- or MyD88-deficient mice) after the injection of Aβ into the striatum compared with wild-type controls. The intra-hippocampal injection of ASC speck into APP/PS1 mice promoted Aβ plaque formation and accumulation. However, it was not possible to induce the spread of Aβ pathology in ASC-deficient APP/PS1 mice [22]. Furthermore, abnormal microglia-specific NLRP3 activation induced chronic neuro-inflammation in the pathological process of AD, resulting in microglia Aβ phagocytic dysfunction, peripheral nerve cell damage, and severe pathological damage [23]. However, this process can be altered by the microglia-specific destruction of the NLRP3 inflammasome. In addition, excessive NLRP3 activation and elevated IL-1β levels in microglia can exacerbate neural tau hyper-phosphorylation, neurofibrillary tangles, and synaptic dysfunction in AD induced by a detrimental chronic inflammatory response [24,25,26,27]. ASC or NLRP3 deficiency was reported to reduce tau pathology and protect against cognitive impairment in tau transgenic mice [28,29]. In addition, IL-1β inhibition in a triple transgenic (3xTg) mouse model of AD induced cognition recovery, tau pathology attenuation, and neuronal beta-catenin pathway function recovery [30,31]. Furthermore, regulation of the NLRP3 inflammasome reduced the inflammatory response, which led to an alleviation of the pathological process of AD. Another study showed that inhibition of the NLRP3 inflammasome reduced Aβ deposition, neuro-inflammation, and cognitive impairment in an AD mouse model [32,33,34,35]. Thus, these findings suggest that Aβ-induced NLRP3 inflammasome activation promotes the pathogenesis of AD by triggering the release of pro-inflammatory cytokines and nerve inflammation. Consistently, brain samples from human AD patients have shown increased caspase-1 activity compared with age-matched control patients without dementia [2].

## 4. Natural Extraction Treatment Studies for NLRP3 Inflammasome Regulation in AD

As mentioned above, microglial NLRP3 inflammasome activation is an important characteristic of AD pathogenesis. Several mouse model studies have shown that NLRP3 or caspase-1 deficiency markedly reduced the Aβ burden and reduced cognitive impairment [36,37]. In addition, IL-1β inhibition significantly reduced brain nerve inflammation, alleviated cognitive impairment, and partially reduced Aβ deposition in 3xTg-AD mice [30,31,38]. These findings suggest that novel therapeutic interventions at the molecular level might modulate AD pathology through the inhibition of NLRP3 inflammasome activation. Therefore, therapeutic strategies of inflammasome constructs or downstream signal regulation might reduce neurological inflammation and slow the progression of AD. Here, we present recent studies of natural extraction-mediated NLRP3 inflammasome regulation (Table 1).

*Epimedii Folium* is the dried leaf of *Epimedium brevicornu* Maxim, and *Curculiginis Rhizoma* is the dried rhizoma of *Curculigo orchioides* Gaertn. In previous studies, *Epimedii Folium* and *Curculiginis Rhizoma* (EX) were shown to possess a wide range of neuro-protective, anti-inflammatory, and antioxidant effects. *Epimedii Folium* and *Curculiginis Rhizoma* reduced the content of pro-inflammatory cytokines including tumor necrosis factor-alpha (TNF-α), IL-1β, and IL-6 in the hippocampus and cortex of Aβ-induced rats. EX also reduced the levels of malondialdehyde and increased superoxide dismutase, catalase, glutathione, and glutathione peroxidase in serum. Furthermore, immunohistochemical analysis showed that EX inhibited the expression of NLRP3 and the phosphorylation of MAPKs, NF-κB, MyD88, and cathepsin B [38].

Virgin coconut oil (VCO) is rich in natural antioxidants that regulate lipid profiles and has anti-inflammatory and antioxidant activity, which might be effective for the treatment and prevention of AD. VCO normalized NLRP3 inflammasome gene expression and oxidative stress compared with an Aβ- and high fat diet HFD-induced AD rat model. Congo Red and Cresyl Violet staining, as well as immunohistochemistry showed that VCO improved hippocampus histological changes by reducing Aβ plaques and phosphorylated tau [34,39].

Extra-virgin olive oil (EVOO) protected mice from Aβ-related pathology by restoring the blood–brain barrier (BBB) function at an early age before pathology onset. Oleocanthal (OC), a phenolic compound abundant in EVOO, restored BBB function, reduced Aβ brain levels, and reduced inflammation, as evidenced by reduced astrogliosis and brain levels of cytokines. OC-rich EVOO restored BBB function and reduced AD-associated pathology by reducing neuroinflammation through the inhibition of NACHT, leucine-rich repeats (LRR), and the NLRP3 inflammasome and inducing autophagy [35].

Lychee seeds contain many polyphenols, including rutin, quercetin, catechin, and proanthocyanidins, which inhibited Aβ-induced apoptosis in neuronal cells and the neuroinflammatory response in microglia via the NF-κB pathway. Lychee seed polyphenols (LSPs) significantly decreased monolayer permeability, inhibited NLRP3 inflammasome activation in Aβ_25–35_-induced bEnd.3 cells, and induced autophagy via the AMPK/mTOR/ULK1 pathway. In addition, LSPs improved spatial learning and memory function, increased expression of tight junction proteins (TJ), and inhibited the expression of NLRP3, caspase-1, IL-1β, and p62 in APP/PS1 mice. In another study, LSPs also suppressed activation of the NLRP3 inflammasome by inhibiting the expression of NLRP3 and ASC, the cleavage of caspase-1, and the release of IL-1β in Aβ_1–42_-induced BV-2 cells and recovered PC-12 cells that were induced by incubated medium from Aβ_1–42_-treated BV-2 cells. Moreover, LSPs improved cognitive function and inhibited NLRP3 inflammasome activation in APP/PS1 mice [40,41].

*Picrorhiza kurroa* Bentham, *Scrophulariae* (PK) is a commonly used herb in Ayurvedic medicine, and several studies reported its therapeutic effects for various diseases related to its antioxidant, anti-inflammatory, anti-allergic, and anti-asthma effects. In a mouse model of AD, PK altered disease-related microglial neuro-inflammation, as evidenced by shifting microglia phenotypes from the inflammatory form to the anti-inflammatory form and inhibiting the nucleotide-binding oligomerization domain, leucine-rich repeat, and NLRP3 inflammasome activity. Moreover, PK induced silent information regulator type 1/peroxisome proliferator-activated receptor-γ signaling, decreasing the expression of β-secretase 1, which is involved in Aβ production [39].

## 5. Role of NLRP3 Inflammasome in PD

Recent studies have also reported a role for the NLRP3 inflammasome in PD. The three features of PD are neuro-inflammation, the gradual loss of dopaminergic neurons in the SNPc and striatum, and the associated accumulation of protein aggregates rich in α-synuclein in the Lewy form of disease [42]. α-synuclein acts as a damage-associated molecular pattern and alters microglial TLR expression. An increase in the number of classically activated microglia prior to neuronal loss was reported in a mouse model of α-synuclein overexpression [43]. In 2018, Gordon et al. observed that the activation of microglial NLRP3 inflammasomes could be triggered by the presence of fibrillar α-synuclein and the absence of α-synuclein-mediated dopaminergic neurons. Tissue samples collected from the substantia nigra of brains from patients with PD showed elevated levels of cleaved caspase-1 and ASC. The expression of NLRP3 in activated microglia was also observed in post-mortem tissue lysates. In addition, in a mouse model of PD, microglial NLRP3 activation by α-synuclein was delayed without mediating pyroptosis but caused potent IL-1β and ASC release [44]. Very recently, Panicker et al. revealed an association between misfolded α-synuclein and the activation of microglial inflammation in PD. Previously, they reported that microglia Fyn kinase was rapidly activated by LPS and TNF-α. However, in vitro and in vivo studies showed that the non-receptor tyrosine kinase, Fyn kinase, regulated the microglial uptake of misfolded α-synuclein and subsequent activation of the NLRP3 inflammasome. Fyn kinase regulates NLRP3 inflammasome activation by regulating the protein kinase C (PKC) δ-dependent nuclear translocation of NF-κB p65, promoting ROS generation, and the introduction of α-synuclein, which promotes subsequent NLRP3 inflammasome activation, thereby regulating the LPS-dependent priming of the microglial NLRP3 inflammasome [45,46]. Overall, α-synuclein aggregation and the microglial NLRP3 inflammasome appear to be involved in a vicious cycle of neuronal cell death mediated via high levels of cytokine production during the onset of PD. This demonstrates that the link between α-synuclein and microglial NLRP3 activation is a source of persistent neuroinflammation that can lead to gradual neuronal loss in PD.

## 6. Natural Extraction Treatment Studies for NLRP3 Inflammasome Regulation in PD

Due to the lack of information on how the NLRP3 structure and its activation directly modulate dopaminergic neuronal loss, therapeutic strategies involving selective microglial NLRP3 modulators that could be used as novel therapeutics for PD treatment are studied less than those for AD. Here, we present the results of recent studies of NLRP3 inflammasome-based therapeutic development of natural extractions for AD (Table 1).

Bushen-Yizhi formula (BSYZ) is a traditional Chinese medicine composed of common *Cnidium* fruit, tree peony bark, ginseng root, *Radix Polygoni Multiflori Preparata*, Barbary wolfberry fruit, and *Fructus Ligustri Lucidi*. BSYZ has extensive neuroprotective properties, including anti-senescence and anti-apoptosis effects, as well as affects alleviating oxidative stress in various AD animal models. A recent study confirmed that BSYZ alleviated motor impairment and dopaminergic neuron degeneration, attenuated microglia activation, inhibited NLPR3 activation, and decreased the levels of inflammatory cytokines in the 1-methyl-4-phenyl-1,2,3,6-tetrahydropyridine (MPTP)-induced mouse brain. Furthermore, BSYZ inhibited NLRP3 activation and IL-1β production of 1-methyl-4-phenyl-pyridinium (MPP+)-stimulated BV-2 microglia cells [47].

*Antrodia camphorata* polysaccharide (ACP) is the main component of the natural polyporaceae Aphididae. ACP, a unique polyporaceae fungus in Taiwan, is composed of a variety of monosaccharides. It is well known that various multiple polysaccharides have anti-inflammatory effects. A recent study showed that ACP inhibited the expression of ROS-NLRP3 induced by 6-OHDA, exerted a protective role in MES23.5 cells, reduced the activation level of ROS-NLRP3 in the substantia nigra–striatum, and improved the exercise capacity of PD mice. Another study confirmed that ACP elevated levels of dopamine in the striatum and decreased the expression of the NLRP3 inflammasome signal in the striatum of 6-OHDA-induced PD mice [48,49].

## 7. Discussion

Neurodegenerative diseases, especially AD and PD, are major medical problems and social issues worldwide. AD and PD account for the largest proportion of neurodegenerative diseases worldwide, and they were the neurological diseases that increased the most between 1990 and 2016 [50]. However, effective treatments that improve the clinical outcomes of patients with AD and PD have not been developed yet.

In recent years, many studies have shown the disease-improving effects of single compounds derived from natural extractions, such as fruits, vegetables, and Chinese herbal medicines, for the treatment of AD and PD, by regulating NLRP3 inflammasome assembly.

Baicalin, a flavonoid compound isolated from the root of *Scutellaria baicalensis* Georgi, has potent anti-inflammatory effects. In APP/PS1 transgenic mice, the oral administration of baicalin for 33 days significantly attenuated spatial memory dysfunction in the passive avoidance test and Morris water maze test. Although the reduction in Aβ deposition was not significant, baicalin reduced microglia activation, IL-18, IL-1β, and iNOS expression in APP/PS1 mice and LPS/Aβ-stimulated BV2 cells. These results indicated the protective effects of baicalin in AD progression were related to the inhibition of microglia-induced neuroinflammation by suppressing the activation of NLRP3 inflammasomes and the TLR4/NF-κB signaling pathway [51].

Flavocoxid, a mixture of purified baicalin and catechin, has anti-inflammatory properties including inhibition of the peroxidase activity of COX enzymes, a common target of non-steroidal anti-inflammatory drugs. After the administration of flavocoxid to 3xTg-AD mice for 3 months, spatial learning and memory were significantly improved in the Morris water maze test, whereas amyloid deposit, eicosanoid production, neuron apoptosis, p-APP, and p-tau were significantly reduced. In addition, flavocoxid reduced Aβ plaques and neuronal loss in the hippocampus and isocortex of 3xTg-AD mice. In that study, the neuroprotective effects of flavocoxid were associated with the regulation of NLRP3 inflammasome-mediated IL-1β production [33].

Dihydromyricetin is an abundant flavanonol in *Ampelopsis*. A previous study reported that supplementation of dihydromyricetin for 4 weeks reduced microglia activation and expression of the NLRP3 inflammasome components, including NLRP3 and caspase-1, and Aβ levels in APP/PS1 mice. Notably, dihydromyricetin significantly enhanced the conversion of BV-2 microglia from the M1 phenotype to the M2 phenotype, which may promote the clearance of Aβ [52].

Resveratrol is a natural plant polyphenol compound abundant in red grape skin and wine. Resveratrol alleviated learning and memory impairments, and increased the expression of Sirt1, AMPK, and PKC-1α in the hippocampus and prefrontal cortex of Aβ1-42-induced AD mice. In this AD animal model, resveratrol significantly reduced Aβ-mediated inflammatory responses by inhibiting NF-𝜅B, IL-1β, and NLRP3 expression [53].

Astaxanthin is a natural carotenoid abundant in marine organisms such as shrimp, crab, krill, salmon, and microalgae. In a previous study, supplementation of 0.2% astaxanthin in the diet of APP/PS1 mice for 60 days enhanced learning and memory in the radial eight-arm maze and Morris water maze tests, and reduced amyloid plaques, hyperphosphorylation of tau, microglia activation, and NLRP3 inflammasome assembly [54].

Tenuigenin (senegenin), a saponin isolated from *Polygala tenuifolia* root, is commonly used in Chinese medicine. Pre-treatment of MPTP-induced PD mice with tenuigenin for 10 days alleviated motor impairments, increased levels of dopamine and its metabolites in the striatum, reduced the loss of tyrosine hydrolase-positive neuronal cells, and attenuated the expressions of IL-1β, caspase-1, and NLRP3 in the substantia nigra. In addition, treatment with tenuigenin reduced intracellular ROS and NLRP3-mediated inflammatory responses in microglia BV-2 cells [55].

Table 2 shows that these single compounds modulate the intracellular signaling cascade of the NLRP3 inflammasome/caspase-1/IL-1β axis in vivo to regulate inflammatory responses, alleviate oxidative stress, and improve symptoms of neurological disorders. However, there have been no clinical studies applying these experimental data. Although inflammasome downstream effectors including anakinra (IL-1 receptor antagonist), canakinumab (IL-1β neutralizing antibody), and rilonacept (soluble decoy receptor for IL-1β and IL-1α) have been approved as therapeutic agents [56], there have been no reports of their use in clinical trials of neurodegenerative diseases including AD and PD.

Therefore, the discovery of inhibitors of NLRP3 inflammasome activation from bioactive compounds derived from natural substances might be a promising future strategy for the prevention and treatment of neurodegenerative diseases. Our study showed that β-carotene, sweroside, and sulforaphane [57,58,59], acting as NLRP3 inhibitors, might be applicable to the prevention of AD and PD.

In addition, knowledge regarding the neuroprotective properties of these bioactive compounds mediated through the regulation of the NLRP3 inflammasome is still lacking. Many candidate substances, such as the single compounds presented in Table 2, have been reported to have therapeutic effects. To better understand the role of the NLRP3 inflammasome in AD and PD, further studies on the activation mechanism of the NLRP3 inflammasome using more sophisticated animal model experiments and molecular-targeted NLRP3 treatments need to be performed.

In conclusion, there is extensive experimental evidence that NLRP3 inhibitors can modulate NLRP3 inflammasome activation to prevent or treat AD and PD; therefore, it would be of great value to discover various NLRP3 modulators and use them to develop new therapeutic strategies for treating neurodegenerative diseases. In addition, clinical studies of potential therapeutic agents are needed to study the detailed mechanisms that ultimately target the NLRP3 inflammasome in AD and PD.

## 8. Conclusions

In conclusion, this review summarizes the current knowledge of NLRP3 inflam-masome activation in AD and PD. The NLRP3 inflammasome serves as an intracellular sensor for host-derived risk signals associated with neurological disorders. Especially, recent studies have shown much evidence that AD and PD are induced through the intracellular signaling cascade of the NLRP3 inflammasome/caspase-1/IL-1β axis. This evidence has shown that NLRP3 inflammation plays a crucial role in the development of AD and PD.

In addition, this review presents evidence that several recent preclinical studies that numerous plant-derived chemicals can exert beneficial effects on AD and PD through various inhibitory functions on NLRP3 inflammasome assembly and activation. For decades, natural products have been regarded as potential substances for the prevention or treatment of cancer, metabolic and neurodegenerative diseases due to their antioxidant and anti-inflammatory activity. Based on the evidence summarized in this review, the use of natural products that act in various stages of NLRP3 signaling could be a pharmacological approach suitable for the management of AD and PD as well as chronic inflammatory conditions.

However, since the etiological cause and mechanism of pathogenesis in AD and PD are not clearly identified, related research is still ongoing. In addition, in vivo models for AD and PD also have not been clearly established yet due to difficulties with proceeding to clinical trials. In particularly, there are currently no clinical trials showing the efficacy of natural products targeting NLRP3 in humans. Moreover, it is also necessary to further understand whether it is possible to inhibit the activation of the NLRP3 inflammasome at the transcriptional or post-transcriptional level or directly modify conformation. Nevertheless, the accumulated evidence has shown that the NLRP3 inflammasome plays an important role in the pathogenesis of various neuroinflammation and neurological disorders [60] and that natural products have an effect in the prevention and treatment of various inflammation prevention and inflammation-derived diseases. Therefore, natural product research is still needed through the regulation of NLRP3 inflammasome assembly in a wide range of neuropathies to identify various pharmacological mechanisms and develop therapeutic agents for such neuropathies.

## Figures and Tables

**Figure 1 ijms-22-01324-f001:**
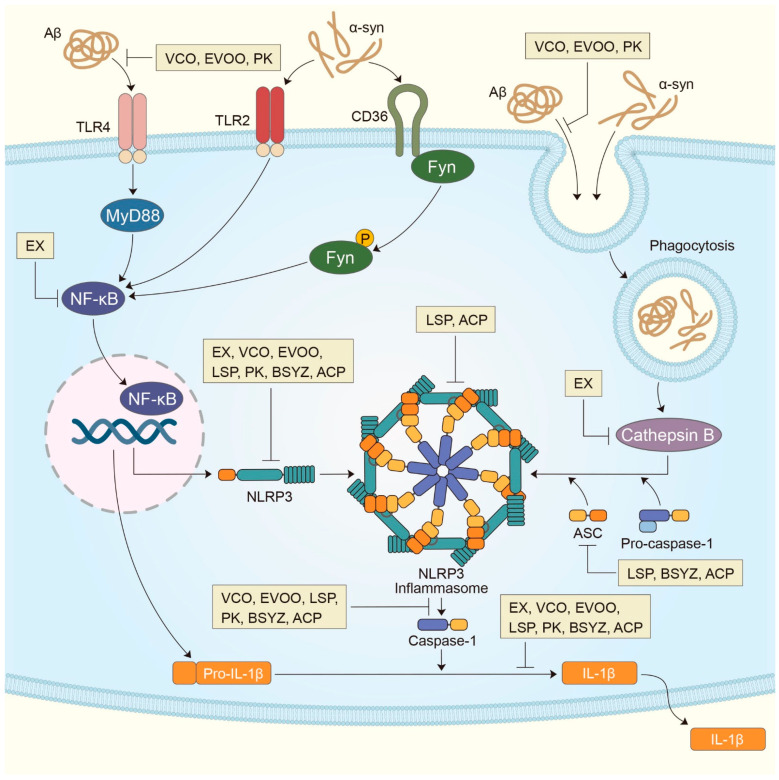
Mechanism of NLRP3 inflammasome activation in AD and PD with potential intervention of natural extractions. In the transcriptional NLRP3 inflammasome activation, Aβ recognized by toll-like receptor 4 (TLR4) in microglia activates the myeloid differentiation primary response protein MyD88 (MYD88)-nuclear-factor-κB (NF-κB) pathway, and α-synuclein recognized by toll-like receptor 2 (TLR4) and CD36 in microglia activates the Fyn kinase-mediated NF-κB pathway to elevate IL-1β and the NLRP3 expression. Along with the expression of NLRP3 following NF-κB translocation, lysosome leakage is induced by the phagocytosis of Aβ and α-syn resulted in the emission of cathepsin B, which in turn induces the NLRP3 inflammasome complex consisting of NLRP3, ASC and pro-casparase-1. The release of IL-1β promoted by active caspase-1 can cause a continuous inflammatory response that contributes to neurodegeneration in AD and PD. Meanwhile, current studies have shown that various natural extractions are able to prevent or treat in AD and PD through regulation of the NLRP3 inflammasome/caspase 1/IL-1β axis. These natural extractions can inhibit NLRP3 inflammasome activation in microglia, thereby the secretion of IL-1β was suppressed. ACP, Antrodia comphorata; BSYZ, Bushen-Yizhi formula; EVOO, Oleocanthal-rich extra-virgin olive oil; EX, Epimedii Folium and Curculiginis Rhizoma; LSP, Lychee seed polyphenol; PK, Picrorhiza kurroa; VCO, Virgin coconut oil.

**Table 1 ijms-22-01324-t001:** NLRP3 inflammasome targeting study of natural extraction in Alzheimer’s disease (AD) and Parkinson’s disease (PD).

Origin of Extraction	Target Indication	Cell or Animal Model	Inducer	Mode of Action and Target Signal	Ref.
Epimedii Folium and Curculiginis Rhizoma	AD	SD rats	Aβ_1–42_	↓ TNF-α, IL-1β, IL-6, MDA, SOD, CAT, GSH, GSH-Px, NLRP-3, ASC, caspase-1, MAPKs, NF-κB, MyD88, cathepsin B ↑ Spatial learning and memory ability	[38]
Virgin coconut oil	AD	Wistar rats	Aβ_1–40_, HFD	↓ TAC, TOS, glutathione, IL-1β, caspase-1, NLRP3, Aβ plaque formation, phosphorylated tau↓ Spatial memory and learning ability, cell density in CA1 region	[34]
Oleocanthal-rich extra-virgin olive oil	AD	TgSwDI mice	Transgenes	↓ Aβ load and plaque, IbaⅠ, MMP9, TRPA-1, protein carbonyl, NLRP3, caspase-1, IL-1β, IL-10 ↑ SOD, autophagy, PSD-95, synapsin-1, SNAP-25	[35]
Lychee seed polyphenol	AD	BV-2, PC-12 cells APP/PS1 mice	Aβ_1–42_ Transgenes	↓ NLRP3, ASC, caspase-1, IL-1β, cell death, apoptosis, NLRP3 inflammasome ↑ Autophagy, cognitive function	[40,41]
		bEnd.3 cells APP/PS1 mice	Aβ_25-35_ Transgenes	↓ NLRP3, caspase-1, ASC, IL-1β, NLRP3 inflammasome ↑ Cell viability and permeability, TJs, autophagy, spatial learning and memory function	
Picrorhiza kurroa	AD	5xFAD mice	Transgenes	↓ Aβ burden, IbaⅠ, IL-1β, NLRP3, caspase-1, C99, BACE1 ↑ Cognitive function, FIZZ1	[39]
Bushen-Yizhi formula	PD	BV-2 cells C57BL/6 mice	MPP^+^ MPTP	↓ Motor impairment, dopaminergic neurons neurodegeneration, NLRP3, IL-1β caspase-1, ASC	[47]
Antrodia comphorata	PD	C57BL/6 mice	6-OHDA	↓ NLRP3 inflammasome ↑ Neurobehavior, dopamine	[48,49]
	PD	MES23.5 cells C57BL/6 mice	6-OHDA	↓ NLRP3, ASC, IL-1β, caspase-1, ROS ↑ Dopaminergic neuron protection	

**Table 2 ijms-22-01324-t002:** NLRP3 inflammasome targeting study of single compound in AD and PD.

Origin of Extraction	Target Indication	Cell or Animal Model	Inducer	Mode of Action and Target Signal	Ref.
Baicalin	AD	SH-SY5Y cells APP/PS1 mice	Aβ_1–42_ Transgenes	↓ Memory impairment, microglial activation, IL-18, iNOS, IL-1β, caspase-3, apoptosis, NLRP3, caspase-1, p-p65, p-IκB, TLR4	[51]
Flavocoxid	AD	3xTg-AD mice	Transgenes	↓ Aβ deposition, PGE-2, LTB-4, NLRP3, IL-1β, p-APP, presenilin-1, p-Tau, p-ERK 1/2, Bax ↑ Cognitive functions, Bcl-2,Bcl-xL	[33]
Dihydromyricetin	AD	BV-2 cells APP/PS1 mice	Aβ_1–42_ Transgenes	↓ Aβ deposition, microglia activation, NLRP3, caspase-1, IL-1β, TNF-α, cognitive deficits ↑ Aβ clearance	[52]
Resveratrol	AD	Kunming mice	Aβ_1-42_	↓ Learning and memory impairments, NF-κB, IL-1β, NLRP3 ↑ SIRT1, AMPK, PGC-1α, neuronal integrity	[53]
Astaxanthin	AD	APP/PS1 mice	Transgene	↓ β deposition, nicastrin, NO, p-tau, microglial activation, IL-1β, TNF-α, Bcl-2, caspase-9, caspase-3, NLRP3, ASC, caspase-1, IL-1β ↑ BACE, SOD, p-GSK3β,	[54,55]
Tenuigenin	PD	BV-2 cells B6 mice	LPS 1-methyl-4-henyl-1,2,3,6-Tetrahydropyridine	↓ NLRP3, IL-1β, caspase-1, ROS ↑ Locomotor activity, dopamine, tyrosine hydroxylase	[55]

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
