# Peer review of "A Novel Treatment Strategy by Natural Products in NLRP3 Inflammasome-Mediated Neuroinflammation in Alzheimer’s and Parkinson’s Disease"

_ijms, 2021, doi:10.3390/ijms22031324_

Round 1

Reviewer 1 Report

COMMENTS TO THE AUTHORS This manuscript reports that Novel Treatment Strategy by Natural Products in NLRP3 Inflammasome-mediated Neuroinflammation in Alzheimer’s and Parkinson’s Disease. The results are quite interesting and scientific but the manuscript needs a revision for the publication in this journal.

Minor concerns

Question 1. You’d be better to suggest the problem or limitation of current treatment strategy/ or drug in AD and PD. It could strengthen the importance of natural products in the treatment of AD and PD.

Reviewer 2 Report

The authors did a large and fairly comprehensive review of the natural product-induced effects on NLRP3 inflammasome on Alzheimer's disease and Parkinson's disease.

The review is well written and easy to read.

I only have a few comments

Lines 112-127, here the authors forget to quote the work of Giuliani et al., 2014, Neurobiology of Aging, where it is point out that IL-1 beta and Caspace-3 are increased in 3x-Tg transgenic mice.

Lines 114-116, the authors wrote “Another study showed that inhibition of the NLRP3 inflammasome reduced Aβ deposition, neuro-inflammation, and cognitive impairment in an AD mouse model [28]. " These results were also obtained by Bitto et al., your Ref. 47. I suggest adding it also after ref. 28.

Conclusion: lines 239-305, why do the authors insert here discussion of other natural compounds instead of talking about them in the above appropriate AD and PD sections ?

Lines 296-297: Why aren't in Table 2 results of the authors' studies 53-55 on beta-carotene, sweroside and sulforaphane?

Reviewer 3 Report

The manuscript deals with the potential of natural products as therapeutic intervention for the prevention or treatment of NLRP3 inflammasome-mediated neurological disorders. The manuscript is well written and follows a good scientific information flow. There are a few modifications that can help the readability. English grammar changes need to be incorporated. The manuscript is well within the prospects of the journal topics and style.

ABSTRACT

Abstract needs grammar correction and also needs some restructuring. A review cannot overcome limitations of experimental studies, rather it can provide a prescriptive or a solution or a basis to explain the limitations. Please rephrase that sentence.

The body of the manuscript needs to be clearly divided into:

INTRODUCTION: ending with the significance of the study and the work aimed in the review

Main idea and then SUb-headings under the main idea.

DISCUSSION

CONCLUSION: especially emphasizing the deductions that closely match the hypothesis and the aim of the article.
